# Treating Neurogenic Lower Urinary Tract Dysfunction in Chronic Spinal Cord Injury Patients—When Intravesical Botox Injection or Urethral Botox Injection Are Indicated

**DOI:** 10.3390/toxins15040288

**Published:** 2023-04-17

**Authors:** Po-Cheng Chen, Kau-Han Lee, Wei-Chia Lee, Ting-Chun Yeh, Yuh-Chen Kuo, Bing-Juin Chiang, Chun-Hou Liao, En Meng, Yao-Lin Kao, Yung-Chin Lee, Hann-Chorng Kuo

**Affiliations:** 1Urologic Department, En Chu Kong Hospital, New Taipei City 237414, Taiwan; 2Division of Urology, Department of Surgery, Chi Mei Medical Center, Tainan 71004, Taiwan; 3Department of Urology, Kaohsiung Chang Gung Memorial Hospital, College of Medicine, Chang Gung University, Taoyuan 33302, Taiwan; 4Division of Urology, Department of Surgery, Taiwan Adventist Hospital, Taipei City 10556, Taiwan; 5Department of Urology, Yangming Branch of Taipei City Hospital, Taipei 11146, Taiwan; 6Department of Exercise and Health Sciences, University of Taipei, Taipei 111036, Taiwan; 7College of Medicine, Fu-Jen Catholic University, New Taipei City 24205, Taiwan; 8Department of Urology, Cardinal Tien Hospital, New Taipei City 23148, Taiwan; 9Divisions of Urology, Department of Surgery, Cardinal Tien Hospital, New Taipei City 23148, Taiwan; 10School of Medicine, Fu Jen Catholic University, New Taipei City 242062, Taiwan; 11Division of Urology, Department of Surgery, Tri-Service General Hospital, National Defense Medical Center, Taipei 11490, Taiwan; 12Department of Urology, National Cheng Kung University Hospital, College of Medicine, National Cheng Kung University, Tainan 704, Taiwan; 13Department of Urology, Kaohsiung Municipal Siaogang Hospital, Kaohsiung 812, Taiwan; 14Department of Urology, Hualien Tzu Chi Hospital, Buddhist Tzu Chi Medical Foundation and Tzu Chi University, Hualien 97004, Taiwan

**Keywords:** spinal cord injury, botulinum toxins, lower urinary tract symptoms

## Abstract

Lower urinary tract symptoms (LUTS), such as urgency, urinary incontinence, and/or difficulty voiding, hamper the quality of life (QoL) of patients with spinal cord injury (SCI). If not managed adequately, urological complications, such as urinary tract infection or renal function deterioration, may further deteriorate the patient’s QoL. Botulinum toxin A (BoNT-A) injection within the detrusor muscle or urethral sphincter yields satisfactory therapeutic effects for treating urinary incontinence or facilitating efficient voiding; however, adverse effects inevitably follow its therapeutic efficacy. It is important to weigh the merits and demerits of BoNT-A injection for LUTS and provide an optimal management strategy for SCI patients. This paper summarizes different aspects of the application of BoNT-A injection for lower urinary tract dysfunctions in SCI patients and provides an overview of the benefits and drawbacks of this treatment.

## 1. Introduction

The annual incidence of traumatic spinal cord injury (SCI) in the United States is around 40 per million, with a reported prevalence of around 721 cases per million people [1]. In Taiwan, the reported annual incidence is about 56.1 per million in Hualien County and 14.6 per million in Taipei [1]. Patients sustaining a traumatic SCI may experience lower urinary tract symptoms (LUTS) of varying severity caused primarily by two mechanisms: failure to store urine and failure to empty the bladder. Conventionally, alpha-blockers were used to improve bladder emptying, and antimuscarinic agents, or beta-3 agonists, were used for detrusor overactivity. However, some SCI patients were dissatisfied with the treatment effects or unable to tolerate the adverse events of oral medications [2,3,4,5]. Botulinum toxin A (BoNT-A) induces muscular paralysis by selectively regulating neurotransmitters (acetylcholine, substance P, and calcitonin gene-related peptides) from motor nerve endings. BoNT-A also could decrease pain and sensory disturbance by modulating sensory receptors and exerting an anti-inflammatory effect [6]. BoNT-A injections into the detrusor muscle, urethral sphincter, or both are used as an effective treatment for SCI patients with LUTS to facilitate urinary control and bladder emptying [6].

BoNT-A is a single-chain polypeptide comprising a 100 kDa heavy chain and a 50 kDa light chain [7]. The heavy chain binds to the nerve terminal, the toxin translocates into the cell cytoplasm through endocytosis, and the light chain cleaves synaptosomal-associated protein-25 (SNAP25), a part of the soluble N-ethylmaleimide-sensitive factor attachment protein receptor (SNARE) protein responsible for assisting the neurotransmitter-containing vesicles to fuse with the nerve terminal membrane, to block the neurotransmission. BoNT-A decreases neurogenic detrusor overactivity (NDO) by obstructing acetylcholine release and inhibiting detrusor contraction [8]. Other probable mechanisms include a decrease in the function of muscarinic receptors or the release of adenosine triphosphate (ATP) to the detrusor muscle from BoNT-A, resulting in decreased involuntary detrusor contraction [8]. The mediating sensory input of the bladder is another possible pathway to improve NDO. Reduction in purinergic receptors (P2X3, P2X2), or transient receptor potential vanilloid subfamily-1 (TRPV1) receptors on the urothelium has been observed in patients who received BoNT-A detrusor injection [8]. A similar mechanism possibly underlies the therapeutic effects of BoNT-A injection in the urethral sphincter for detrusor sphincter dyssynergia (DSD). Inhibiting neurotransmission to the urethral sphincter muscle can temporarily relax the sphincteric muscles to improve efficient urination in patients with both neurogenic and nonneurogenic voiding dysfunction [8].

Several commercial formulations of BoNT-A, namely onabotulinum toxin A, abobotulinum toxin A, incobotulinum toxin A, and rimabotulinum toxin B, are available [9] with varying potency [10]. Onatotulinum toxin A (trade name: “Botox”) has been approved by the U.S. Food and Drug Administration for NDO and is widely used in clinical research [9]. The potency of 200 U of onabotulinum toxin A is equivalent to around 600–800 U of abobotulinum toxin A [11,12], and switching between formulations may be useful when one formulation is not clinically effective [10]. According to a systematic review, abobotulinum toxin A was found to be more effective than onabotulinum toxin A in reducing urge incontinence [12]. For the current review, we have used the dosage reported for Botox injections in the literature.

## 2. BoNT-A Detrusor Injection for Neurogenic Detrusor Overactivity and Urinary Incontinence in Chronic SCI Patients

BoNT-A was initially introduced in urology for treating patients with neurogenic lower urinary tract dysfunction (NLUTD) due to chronic SCI [13,14]. Detrusor BoNT-A injection reduces the incidence of NDO and lowers intravesical pressure, which improves urinary incontinence [15,16,17]. Subsequent studies proved the efficacy of BoNT-A on autonomic dysreflexia (AD) in patients with high-level SCI, in addition to treating NDO and urinary incontinence in pediatric patients with myelomeningocele or other NLUTDs [6].

The U.S. Food and Drug Administration and several European countries approved a 200 U BoNT-A detrusor injection as the standard treatment dose for urinary incontinence caused by NDO in SCI patients [11,12,18]; however, preliminary studies used 300 U to demonstrate its therapeutic effect of increasing maximal bladder capacity and compliance and decreasing reflux volume [19,20,21,22]. Compared with placebo, detrusor BoNT-A injection was effective in reducing NDO caused by SCI or multiple sclerosis [23]. Eventually, researchers compared the therapeutic efficacy of the 200 U and 300 U dosages and found that the former achieves comparable effects with possibly fewer adverse events (AEs), leading to the adoption of the 200 U dose by urologists [18]. Furthermore, repeat BoNT-A injections were found to have similar efficacy to the first one [24]. A pooled data analysis by Mangera et al. revealed detrusor BoNT-A injection in SCI patients could facilitate a decrease in the daily incontinence episodes (63%), the frequency of clean intermittent catheterization (CIC) (18%), and maximal detrusor pressure (42%) as well as an increase in the cystometric bladder capacity (68%) and reflux volume (61%) [6,25]. Detrusor BoNT-A injection can also significantly improve health-related quality of life (QoL) indexes [6,26] and decrease the rate of symptomatic urinary tract infection (UTI) in SCI patients [6,27]. The detrusor BoNT-A injection-induced improvement in bladder compliance and urinary continence can persist for up to nine months [16,28], and repeated injections can sustain improvement in continence and QoL [29].

However, symptom evaluation alone may not be sufficient. High intravesical pressure with potential upper urinary tract damage might be missed in some patients with urinary continence after detrusor BoNT-A injection [30]; therefore, a urodynamic examination might be necessary for some patients even with obvious symptomatic relief after detrusor BoNT-A injection [30]. Besides, detrusor BoNT-A injection usually impairs detrusor contractility, leading to a large postvoiding residual volume (PVR) or urinary retention in patients with NDO; hence, patients receiving detrusor BoNT-A injection may require CIC for large PVR and suffer from subsequent UTIs [15].

Several studies compared different BoNT-A injection methods for patients with NLUTD. Krhut et al. compared BoNT-A detrusor injection with submucosal injection in SCI patients with NDO and found no significant difference between the two groups regarding symptomatic improvement, change in urodynamic parameters, efficacy duration, and AEs [31]. Samal et al. also reported concurring findings as the two methods gave similar results in terms of urgency incontinence episodes, number of catheterizations, urodynamic profile, and efficacy duration [32]. Interestingly, Taha et al. compared a trigone-including injection with a trigone-sparing injection in SCI patients with refractory NDO and found that the former group had a significantly lower incontinence rate, higher complete dryness rate, and higher reflux volume [33]. No injection-related vesicoureteral reflux was found in either group. Another randomized control trial compared combined BoNT-A 160 U trigone-sparing and 40 U trigone-including injections with 200 U trigone-sparing BoNT-A injection for NDO in SCI patients [34]. Including the trigone resulted in superior outcomes for QoL, mean urinary incontinence episodes, complete dryness rate, mean voided volume, detrusor pressure, and involuntary detrusor contraction without any vesicoureteral reflux. In another randomized control trial, the effectiveness of combined BoNT-A injection (240 U into the detrusor and 60 U into the trigone) was compared to that of nontrigonal injection (300 U into the detrusor but not the trigone). The trial found that both methods had similar results, but the trigonal injection was found to be safer and more effective than the nontrigonal injection and did not increase the rate of vesicoureteral reflux [35]. The trigone has abundant sensory neural fibers and is highly sensitive to pressure changes, which possibly influences detrusor overactivity; therefore, the denervating effect of the trigone-including injection is more effective in inhibiting involuntary bladder contraction [36].

Treatment efficacy is a crucial factor for deciding whether SCI patients should receive a detrusor BoNT-A injection [37]. Certain pretreatment factors like higher maximum detrusor pressure and lower bladder compliance. Lower maximal cystometric capacity, and poor response after the first injection have been associated with higher treatment failure rates [38,39,40]. NDO patients with initially higher incontinence reduction rates after the first detrusor BoNT-A injection showed better outcomes for incontinence reduction and QoL during the subsequent treatment period [41]. Some NDO patients with poor response to the first dosage also reported gradually higher incontinence reduction after receiving subsequent BoNT-A therapy. Therefore, a second dose of detrusor BoNT-A injection must be given to patients who did not show a good response to the first injection before classifying them as poor responders.

## 3. Urethral Versus Detrusor BoNT-A Injection in SCI Patients with Urinary Incontinence and Incomplete Bladder Emptying

The presentation of NLUTD depends on the level of SCI—patients with a suprasacral cord injury might develop NDO with or without DSD, and symptoms of frequency, urgency, with or without urgency urinary incontinence, and incomplete voiding [42]. Patients with a sacral lesion might present with a poorly contracting detrusor with incomplete voiding and/or stress-related urinary incontinence associated with a weak sphincter, while a cauda equina lesion may develop detrusor areflexia and incompetent sphincter relaxation [42]. SCI patients commonly have both voiding and storage symptoms [43]; targeting the detrusor for urinary incontinence might worsen bladder emptying ability, whereas targeting the urethral sphincter for voiding dysfunction might aggravate the incontinence. It is a challenge for both the urologist and the patient to balance between being dry and complete bladder emptying.

While a detrusor BoNT-A injection of 200 U or 300 U can improve bladder compliance and restore urinary continence in SCI patients with NDO [16], this treatment usually hampers detrusor contractility. A large PVR or urinary retention might become bothersome, and about 70% of patients require periodic CIC and develop subsequent UTIs [15]. Therefore, a lower dose (200 U over 300 U) of BoNT-A is sufficient to produce similar improvements in urinary incontinence and urgency episodes without altering the spontaneous voiding function that is required [44,45].

Dykstra et al. first reported the use of urethral sphincter BoNT-A injection to improve bladder emptying in SCI patients with DSD [46]. This injection can decrease the mean maximal urethral pressure, duration of DSD, and PVR for three to nine months [13]. Improved voiding and reduced CIC frequency also improve the QoL [2]. However, a urethral sphincter injection would increase the severity of urinary incontinence and urgency or urge urinary incontinence episodes, which might lead to patient dissatisfaction [2,47]. Exacerbated incontinence and urgency episodes might compel the patient to select detrusor BoNT-A injection subsequently [48,49].

The optimal condition for chronic SCI patients is self-voiding without urinary incontinence or severe urgency. In order to achieve this goal of treatment result, concomitant BoNT-A detrusor and urethral sphincter injections might be adopted in patients craving less urinary incontinence and preservation of spontaneous voiding. Huang et al. reported that combined BoNT-A (200 U) detrusor and (100 U) urethral sphincter injections might induce a significant reduction in both detrusor and urethral pressure without increasing PVR, daily CIC frequency, or daily pad use [50]. Another study with the same BoNT-A dosage revealed that a combined injection could lower the detrusor and urethral pressures to improve the patient’s QoL while protecting the upper urinary tract [51].

## 4. Long-Term Adherence to BoNT-A Injection and Patient Satisfaction

Sex-based differences are an important concern while treating patients with NDO or DSD. Male SCI patients can use an external urine collection device to prevent soiling and usually can do without being totally dry. On the other hand, female SCI patients tend to use a diaper and would rather prefer to be totally dry; if possible, be free of diapers and decrease the need for CIC to a minimum. A small dose of BoNT-A detrusor injection is adequate to increase bladder capacity and decrease urinary incontinence in this condition; Around 24% of patients can void by abdominal tapping (tapping the suprapubic area to induce reflex contraction of the bladder) without requiring CIC [52,53]. A dose–response study compared BoNT-A injections of 200 U, 100 U, and 50 U with placebo injections into the detrusor muscle [13]. The study found that the 50 U dose did not show any significant improvement over the placebo for any of the efficacy parameters. However, the 100 U dose showed some improvement over the placebo, although the observed magnitude of change was generally less favorable compared to that seen with the 200 U dose [13].

Before administering a BoNT-A detrusor injection in SCI patients, the following issues need to be considered:Behavioral modifications should be the primary strategy.BoNT-A injections could be considered in the case of failure of other treatments or intolerable adverse effects of oral medications, such as antimuscarinics or beta-3 agonists.Most patients require CIC after BoNT-A detrusor injection. Patients unable to perform CIC might not be suitable for this treatment.Regular monitoring is essential to check upper urinary tract function and prevent adverse effects and the occurrence of UTIs and large PVR.Repeated injections are required to maintain therapeutic efficacy in SCI patients.

Repeat detrusor injections are required to sustain the therapeutic efficacy of BoNT-A on NDO. An estimated 67% of SCI patients, who received BoNT-A injections for NDO, continued with repeat injections during a five-year follow-up [54]. Of these patients, 90% reported high satisfaction and were willing to consider periodic BoNT-A injections as a long-term treatment option [6,17,54]. Herbert et al. conducted a retrospective chart review for SCI patients with NDO who received BoNT-A therapy and reported a 59.1% adherence rate at 5 years and 50% at 10 years [40]. Baron et al. also described similar adherence rates (5 years: 63.9%; 10 years: 49.1%) [55]. The most common reasons for discontinuation were lack of efficacy and AEs, such as UTI, urinary retention, and hematuria [37].

Adherence to detrusor BoNT-A injections in SCI patients is highly associated with the treatment outcome [54]. Repeated injections allow the maintenance of favorable effects and hence, greater patient satisfaction [37,54]. However, SCI patients with voiding dysfunction are not equally satisfied with long-term urethral sphincter BoNT-A injections as they are with detrusor injections [3]. Only 35.6% of patients in this study continued urethral sphincter BoNT-A injection therapy during a median follow-up of five years [3]. The low satisfaction rates after urethral sphincter injection were mainly caused by a high incontinence grade, inefficient therapeutic efficacy, and failure to help them wean off of CIC [3].

## 5. Adverse Events Related to BoNT-A Injections

The most common AEs with BoNT-A detrusor injections are symptomatic UTI, urinary retention, and hematuria [4]. BoNT-A detrusor injections are effective in reducing detrusor pressure and urinary incontinence and increasing maximal bladder capacity [4]. However, they also impair detrusor contractility leading to subsequent large PVR or urinary retention. Although BoNT-A urethral sphincter injection can facilitate bladder emptying and increase the flow rate (duration around three to nine months), it also enhances the risk of urinary incontinence, urgency sensation, and de novo frequency [3]; seldom, patients may be disappointed by the AEs that exceed the therapeutic effect. Largely, dissatisfaction with BoNT-A urethral sphincter injection is rooted in amplified urinary incontinence, whereas larger PVR requiring CIC is the primary reason for discontent with detrusor injection [52].

Intolerable AEs often cause patients to forsake repeat BoNT-A injections, although these patients can achieve complete dryness after detrusor injections. Chen et al. retrospectively reviewed 223 patients with chronic SCI who received BoNT-A detrusor injections for NDO and urinary incontinence; only 108 patients (48.4%) continued with repeat injections during the mean ten-year follow-up [17]. Among those discontinuing BoNT-A therapy, 41 patients (46.6%) discontinued due to UTIs, while 15 patients (17%) deferred due to the burden of CIC. Likewise, Hebert et al. reviewed 128 SCI patients receiving repeated detrusor BoNT-A injections for NDO, of which 58 discontinued therapy over a median follow-up of ten years [40]. Seventeen patients stopped therapy due to AEs despite the therapeutic efficacy of detrusor BoNT-A injection. Table 1 summarizes the therapeutic effects and AEs of the detrusor and urethral sphincter BoNT-A injection for patients with chronic SCI with NDO and/or DSD.

## 6. Active Management of Chronic SCI Patients with Autonomic Dysreflexia

AD is an acute systemic disorienting autonomic response to specific stimuli, which may occasionally be potentially fatal. It typically develops in a complete SCI above the T6 level [56] and usually presents with pounding headaches, severe paroxysmal hypertension, bradycardia, flushing and sweating of the face and body above the level of the lesion, nasal congestion, blurred vision, and a sense of apprehension or anxiety. Some patients show irritability, combative behavior, or cognitive impairment [57]. AD commonly develops from a stimulus of bladder distention and stool impaction. UTI is also a potential cause of AD in SCI patients regardless of an indwelling urinary catheter [58]. In SCI, the reflex activity of the sympathetic nervous system, which responds to sensory stimuli, cannot be controlled by the brainstem. Therefore, an increase in blood pressure is maintained via sympathetic vasoconstriction; the baroreceptors sense the rise in blood pressure and trigger parasympathetic activity, which results in compensatory bradycardia [58].

Ke et al. reported that symptomatic AD resolved in 88.2% of SCI patients who underwent transurethral incision at the bladder neck (TUI-BN) with a significant increase in peak flow rate and decreased PVR [59]. Probably, sympathetic activity decreases either during bladder distention or when initiating voiding, which forms the therapeutic mechanism of the resolution of AD after TUI-BN. The authors also observed that 13 patients with impaired detrusor contractility at baseline retained effective and sustained detrusor contraction postoperatively after TUI-BN. An overactive sympathetic response might suppress detrusor contraction and result in low detrusor contractility and incomplete bladder emptying. TUI-BN might interrupt the sympathetic reflex circuit and activity and help these patients regain detrusor contractility.

BoNT-A detrusor or urethral sphincter injections are effective in ameliorating the severity and frequency of symptomatic AD in SCI patients [16,60,61,62]. BoNT-A injection in the detrusor or urethral sphincter possibly reduces the detrusor pressure during the bladder filling or voiding phase to reduce AD; additionally, the detrusor injection might inhibit the sensory afferent pathways that trigger AD. Walter et al. reported that AD severity decreased in 82% of SCI patients after BoNT-A detrusor injection, in addition to a reduction in both the total and bladder-related AD episodes [62]. Fougere et al. also reported that 59% of SCI patients receiving detrusor BoNT-A injections no longer experienced symptomatic AD, and the remaining 41% perceived a significant decrease in the severity [61]. The injections also reduced bladder-related AD events and improved AD-related QoL. A few case studies have also reported the role of urethral sphincter injection in reducing the degree and frequency of AD [60,63]. Therefore, BoNT-A injection (detrusor or urethral sphincter) is a viable option for effective AD control in SCI patients with refractory medical control, such as a selective alpha-1 blocker or a calcium channel blocker.

In contrast, AD might also occur during BoNT-A detrusor injection [4,48]. A study reported that 3.74% of SCI patients who received BoNT-A detrusor injections experienced AD compared to only 0.53% of patients in the placebo group [4]. Exacerbation of AD may result from injection-related bladder wall trauma, which induces acute suburothelial nerve plexus inflammation and sympathetic response activation.

## 7. Augmentation Enterocystoplasty Versus BoNT-A Detrusor Injection in SCI Patients

Augmentation enterocystoplasty (AEC) is an invasive surgery performed to increase bladder capacity and decrease detrusor pressure [64,65]. The bladder is opened, and an anastomosis is made to a detubularized bowel segment; the ileum is the most commonly used site [64,65]. Favorable outcomes have been reported for AEC in terms of improvement in the QoL, renal function preservation, and vesical ureteral reflux resolution in the neurogenic bladder [66]. Additionally, significant improvements in several urodynamic parameters, including increased bladder capacity, decreased maximum detrusor pressure, and detrusor overactivity, have been observed. Most people can become continent postoperatively, while the rate of CIC also increases after surgery [66]. A study reported high satisfaction (96.2%) in SCI patients undergoing augmentation ileocystoplasty, all of whom became completely continent [67].

However, there are certain contraindications for AEC, such as renal insufficiency, inflammatory bowel disease, congenital bowel anomalies, radiation enteritis, short bowel or previous bowel resection, bladder cancer, and inability to perform CIC postoperatively [65,68]. The reported mortality rate for AEC is around 0–3.2% [68]. Early postoperative complications include prolonged ileus, urinary fistula, wound infection, and bleeding requiring reoperation, whereas bacteriuria, UTI, bladder and kidney stone diseases, metabolic disturbances, vitamin B-12 deficiency, bowel complications, and bladder perforation are possible long-term complications [65,68]. Notably, AEC was also associated with an increased risk of malignancy [65,68]. Compared to AE, BoNT-A injection is less invasive with fewer complications and is reversible and easy to perform. In the United Kingdom, the number of AEC procedures decreased from 192 cases in 2000 to 120 in 2010, while that of BoNT-A injections increased from 50 in 2000 to 4088 cases in 2010 [57]. AEC was considered in chronic SCI patients only when conservative treatment, including BoNT-A injection, failed.

Padmanabhan et al. conducted a five-year cost analysis to compare BoNT-A detrusor injection with AEC [69]. Repeat BoNT-A injections are required to maintain clinical efficacy, while AEC is associated with higher long-term complication rates. The cost-analysis model revealed that BoNT-A injections were less when the injection duration was more than 5.1 months, and AEC was cheaper when the complication rate was <14% [69]. However, AEC was typically performed after failure of BoNT-A injection. Furthermore, a cross-sectional study compared the QoL in SCI patients who underwent AEC or BoNT-A detrusor injection and found that continence control and QoL scores were higher in the AEC group [70]. It is possible that the increase in incontinence episodes between consecutive BoNT-A detrusor injections may have caused this difference; accordingly, better re-injection timing might increase the QoL in SCI patients receiving BoNT-A injections [70].

BoNT-A detrusor injection is also a treatment of choice in patients with refractory symptoms after AEC. Toia et al. reported that 43% of SCI patients with failed AEC continued to receive regular BoNT-A detrusor injections (in the remnant native bladder avoiding trigone) with satisfactory symptomatic improvement [71]. Another study reported that 86% of patients with refractory urinary symptoms post-AEC had subjective improvement after receiving BoN-TA injections [72]. In this study, patients received either a detrusor-only injection or a combined detrusor + intestinal part injection, among which 80% of the former group and 91% of the latter reported subjective improvement.

## 8. BoNT-A Injection for Pediatric SCI Patients

Several studies have proved the clinical efficacy of BoNT-A detrusor injection for refractory pediatric neurogenic detrusor overactivity caused by congenital spinal dysraphism, such as myelomeningocele [24,73,74,75,76,77,78]. BoNT-A detrusor injection can also significantly reduce urinary incontinence and detrusor pressure and increase maximal bladder capacity and compliance in pediatric patients [24,73,74,75,76,77,78]. In a study including only pediatric SCI patients, BoNT-A detrusor injection significantly improved the clinical symptoms (decreased urge incontinence episodes and increased dryness rate) and urodynamic parameters (reduced detrusor pressure and duration of first detrusor overactivity during the study) [78]. Detrusor injection of 200 U was shown to be more effective. However, it is advised to avoid exceeding a dosage of 6 U/kg [78]. Furthermore, a trigone-including BoNT-A injection was effective without causing VUR. Concomitant detrusor and urethral sphincter injections have also been used in pediatric patients of NDO with DSD [76]. Compared to a detrusor-only injection, the combined injection can achieve comparable improvements in detrusor pressure, bladder capacity, and incontinence rate, along with significantly reduced PVR. Hence, the combined injection may be useful in pediatric NDO patients with DSD requiring a reduction in PVR. Repeated BoNT-A injections also demonstrated a persisting effect in the pediatric neurogenic bladder, with the treatment effect lasting for nine months in 84% of patients during a median follow-up of 4.5 years [18,66]. Low-compliant bladder and poor response to the first BoNT-A injection were risk factors for treatment failure [74,79].

## 9. Conclusions

Managing NLUTDs in patients with chronic SCI requires deliberating over several aspects of the condition, such as the level and severity of SCI, hand dexterity and walking ability, vocation and daily routine of the patient, abdominal muscle function and bladder sensation, clinical symptoms of voiding difficulty and urinary incontinence, bladder pressure, and upper urinary tract condition. Clinicians must review the pros and cons of the BoNT-A therapy with the patients before deciding on the site of injection (detrusor and/or urethral sphincter) to address their bladder emptying or storage problems. Regular urodynamic study and LUTS evaluation are essential for optimal adjustment of dosage and duration between BoNT-A injections. BoNT-A injections are a powerful tool for treating refractory NLUTDs in SCI patients. Despite the potential AEs, the effects typically wear off after six to nine months. Compared to other invasive treatments, the reversibility of BoNT-A injections is one of its major advantages.

## Figures and Tables

**Table 1 toxins-15-00288-t001:** A summary of therapeutic effects and adverse events related to detrusor and urethral BoNT-A injection in chronic SCI patients with NDO and/or DSD [2,3,4,5].

	Therapeutic Efficacy	Adverse Events
Detrusor injection	Maximal bladder capacity ↑	Post-void residual volume ↑
	Voided volume ↑	Maximum flow rate ↓
	Detrusor pressure ↓	Symptomatic UTI ↑
	Incidence of DO ↓	Urinary retention rate ↑
	Involuntary detrusor contraction ↓	Hematuria ↑
	Urinary incontinence rate ↓	
	Diaper use ↓	
Urethral sphincter injection	Urethral pressure ↓	Urinary incontinence ↑
	Post-void residual volume ↓	Urgency episode ↑
	Maximum flow rate ↑	De novo frequency ↑
	Detrusor pressure ↓	
	Catheterization rate ↓	

↓: decrease; ↑: increase; DO: detrusor overactivity; UTI: urinary tract infection.

## Data Availability

The data presented in this study are available this article.

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
