# Peer review of "Treating Neurogenic Lower Urinary Tract Dysfunction in Chronic Spinal Cord Injury Patients—When Intravesical Botox Injection or Urethral Botox Injection Are Indicated"

_toxins, 2023, doi:10.3390/toxins15040288_

Round 1

Reviewer 1 Report

General Comments

This review article deals with a timely topic of using Botox injections for the treatment of neurogenic lower urinary tract dysfunction in chronic SCI patients.  The article is generally well written with some exceptions (see below), and major revisions are recommended before publication.

1. The title itself feels like a false dichotomy – the site of injections for Botox are considered by clinicians based on the symptoms and need of the patient, as suggested by the authors themselves. Therefore, “Intravesical Botox Injection or Urethral Botox Injection?” seems like a “false” dichotomy that isn’t there.

2. In general, there are too many acronyms. Please reduce their number, if possible, to aid with the readability of the article. Also add a list of acronyms and their definitions.  

3. Line 144 – SCI patients do not normally have spontaneous voiding function – and this sentence needs editing since in its current form, its meaning is very unclear.

4. In general, what is the prevalence of “enterocystoplasty” in patients with SCI?  This section needs summary paragraph.

5. The “Conclusions” section is very superficial and does not have synthesized statements or future direction but only a summary of what was said already. It is recommended that authors “conclude” as to what the current state of Botox usage for “Treating Neurogenic Lower Urinary Tract Dysfunction in Chronic Spinal Cord Injury Patients” as suggested by the title.

Specific Comments

1.     Key contributions statement should have a full stop

2.     Page 1, line 30. Need a reference or references after the sentence talking about the adverse effects of oral medications.

3.     Line 171: what is abdominal tapping? May want to describe this a bit more in detail. Is this still prevalent practice? Seems like an outdated practice.

4.     Line 175 – 176 – This is the current recommendation but what is the rationale for this? Are there projected side effects for using BoNT-A injections earlier?

5.     Line 185 – Need space after ref #47

6.     Line 228 – “massive disoriented autonomic response” is a very non-specific description of AD. What does massive refer to? What does “disoriented” mean? Do the authors mean “disorienting”? Need more precise language here.

7.     Line 239, in describing AD, it will be useful for readers to know.

8.     Line 267 – Suggest alternative ways of applying BoNT-A to patients to reduce AD episodes during detrusor injections? Are there alternative methods to reduce occurrences of AD in patients with SCI?

Author Response

  1. The title itself feels like a false dichotomy – the site of injections for Botox are considered by clinicians based on the symptoms and need of the patient, as suggested by the authors themselves. Therefore, “Intravesical Botox Injection or Urethral Botox Injection?” seems like a “false” dichotomy that isn’t there.

I changed the topic to “Treating Neurogenic Lower Urinary Tract Dysfunction in Chronic Spinal Cord Injury Patients – When Intravesical Botox Injection or Urethral Botox Injection Are Indicated” The topic was also suggested by other reviewers.

  1. In general, there are too many acronyms. Please reduce their number, if possible, to aid with the readability of the article. Also add a list of acronyms and their definitions.

I deceased the acronyms and add a list of acronyms.

  1. Line 144 – SCI patients do not normally have spontaneous voiding function – and this sentence needs editing since in its current form, its meaning is very unclear.

I can’t find this sentence. Would you please remind me where the sentence is?

  1. In general, what is the prevalence of “enterocystoplasty” in patients with SCI?  This section needs summary paragraph.

I searched the online database and I couldn’t find the references about the prevalence of enterocystoplaty. Reference 60 mentioned, in the United Kingdom, the number of enteroscystoplasty decreased from 192 cases in 2000 to 120 in 2010, while that of BoNT-A injections increased from 50 in 2000 to 4088 cases in 2010. In our another article (toxin2022 Jan; 14(1): 30., doi: 10.3390/toxins14010030), we followed our 118 SCI patients who initially underwent botulinum injection. 5 of them underwent enterocystoplasty subsequently. I think the prevalence of enteroscystoplasty is not high and it decreases gradually.

  1. The “Conclusions” section is very superficial and does not have synthesized statements or future direction but only a summary of what was said already. It is recommended that authors “conclude” as to what the current state of Botox usage for “Treating Neurogenic Lower Urinary Tract Dysfunction in Chronic Spinal Cord Injury Patients” as suggested by the title.

Thank you for your suggestion. I have changed my conclusion.

Specific Comments

  1. Key contributions statement should have a full stop
  2. Page 1, line 30. Need a reference or references after the sentence talking about the adverse effects of oral medications.
  3. Line 171: what is abdominal tapping? May want to describe this a bit more in detail. Is this still prevalent practice? Seems like an outdated practice.
  4. Line 175 – 176 – This is the current recommendation but what is the rationale for this? Are there projected side effects for using BoNT-A injections earlier?
  5. Line 185 – Need space after ref #47
  6. Line 228 – “massive disoriented autonomic response” is a very non-specific description of AD. What does massive refer to? What does “disoriented” mean? Do the authors mean “disorienting”? Need more precise language here.
  7. Line 239, in describing AD, it will be useful for readers to know.
  8. Line 267 – Suggest alternative ways of applying BoNT-A to patients to reduce AD episodes during detrusor injections? Are there alternative methods to reduce occurrences of AD in patients with SCI?

I have corrected my articles according to the suggestion above. Thank you very much for your guidance.

Reviewer 2 Report

Overall and excellent review

- (line 92)There is no guideline statement recommending regular UDS after Botox in patients with NDO.  Unless there is known decreased bladder compliance and the Botox is being utilized to decrease bladder pressures, guidelines do not recommend routine UDS in these patients.

- (line 175) Botox should be offered to patients as an option even if they are tolerating medical therapy as it may offer improvements of QoL.  Patients need not fail oral medication or not tolerate to be offered botox.

-(line 201). Should include rates of retention with each dose of botox here.

-(line 229). This should say symptoms include (not usually present with) as many patients have silent AD and few patients have all of the typical symptoms.  The rest of this section should include reduction in symptomatic AD (again remembering that a portion of AD patients have no symptoms)

- (line 280) include renal insufficiency as a contraindication to augmentation

- (line 294) comparing cost of Botox vs. AEC is not clinically relevant in most cases as AEC is typically only done now for patients who fail botox.

Author Response

- (line 92)There is no guideline statement recommending regular UDS after Botox in patients with NDO.  Unless there is known decreased bladder compliance and the Botox is being utilized to decrease bladder pressures, guidelines do not recommend routine UDS in these patients.

Since reference 24 mentioned that some patients may have injuries to their upper urinary tract system without apparent symptoms. UDS might be helpful for these patients. I have modified my statement.

- (line 175) Botox should be offered to patients as an option even if they are tolerating medical therapy as it may offer improvements of QoL.  Patients need not fail oral medication or not tolerate to be offered botox.

I have modified my statement. Thank you for your guidance

-(line 201). Should include rates of retention with each dose of botox here.

The reference didn’t mention the retention rate. It only mentioned the 38.1 % patients reported not improved and 33.3% patient reported mild improved after botox urethral injection.

-(line 229). This should say symptoms include (not usually present with) as many patients have silent AD and few patients have all of the typical symptoms.  The rest of this section should include reduction in symptomatic AD (again remembering that a portion of AD patients have no symptoms)

I have modified my statement. Thank you for your guidance

- (line 280) include renal insufficiency as a contraindication to augmentation

I have modified my statement. Thank you for your guidance

- (line 294) comparing cost of Botox vs. AEC is not clinically relevant in most cases as AEC is typically only done now for patients who fail botox.

I have modified my statement. Thank you for your guidance

Reviewer 3 Report

This is a well written review about botulinum toxin in NDO patients.

 The title needs to be fine tuned. Bladder and urethral injections have different objectives which in most cases are not alternatives. The title suggest that they may be, which is not the case. Please change to  "When intravesical....or urethral.... are indicated”

 In introduction, line 11 lacks an and instead of a to: . ….from motor nerve endings and to decrease .. …

 In the 2nd paragraph of page 2:

 Abobotulinum toxin was investigated in the phase 3 CONTENT trials and the pool analysis was published in Eur. Urol. in 2022.

The equivalence put in the manuscript turns out to be incorrect since Abobot A 600 to 800 U provided results equivalent to OnabotA 200. Please remove it and describe the data of AbobotA and state that Dysport already received a positive outlook from several European Countries.

 Please quote the following two papers :

 Kennelly et al, Eur Urol. . 2022 Aug;82(2):223-232. doi: 10.1016/j.eururo.2022.03.010.

Cruz F, et al, J Med Econ. 2023 Jan-Dec;26(1):200-207. doi: 10.1080/13696998.2023.2165366

In section 2:

 Quote Kennelly et al, Eur Urol. . 2022 Aug;82(2):223-232. doi: 10.1016/j.eururo.2022.03.010.

 The approval of 200 U Botox is not only in the US, please correct the sentence

 Describe the CONTENT trial and the comparison between the two toxins from Cruz F, et al,  J Med Econ. 2023 Jan-Dec;26(1):200-207. doi: 10.1080/13696998.2023.2165366

In page 3, About trigonal injections in NDO patients please comment the paper:

Chen et al, J Spinal Cord Med. 2020

Concerning urethral injections:

Describe continence and pad usage. The study quoted is small and how many  patienst were dry? Please describe and make a comment. Also describe the duration of effect of urethral injections.

 In section 4 when addressing ideal doses make clear that doses different than 200 for Botox are not licenced. Also leave clear that doses below 100 U are not effective in NDO/SCI patients.  Refer to  Apostolidis World J Urol. 2013

In section 5, comment the duration of effect of BoNT/A injections in the sphincter in the text and in Table 1

 In section 8: It is important to refer the maximal  doses of OnabotA and AbobotA in the paediatric population. And to state the best dose in the Austin study (ref 74)

State if there is any study with sphincter injection in NDO children

Author Response

This is a well written review about botulinum toxin in NDO patients.

 The title needs to be fine tuned. Bladder and urethral injections have different objectives which in most cases are not alternatives. The title suggest that they may be, which is not the case. Please change to  "When intravesical....or urethral.... are indicated”

Thank you for your suggestion; I have made modifications to the topic

 In introduction, line 11 lacks an and instead of a to: . ….from motor nerve endings and to decrease .. …

Thank you for your suggestion; I have made modifications to the sentence.

 In the 2nd paragraph of page 2:

 Abobotulinum toxin was investigated in the phase 3 CONTENT trials and the pool analysis was published in Eur. Urol. in 2022.

The equivalence put in the manuscript turns out to be incorrect since Abobot A 600 to 800 U provided results equivalent to OnabotA 200. Please remove it and describe the data of AbobotA and state that Dysport already received a positive outlook from several European Countries.

 Please quote the following two papers :

 Kennelly et al, Eur Urol. . 2022 Aug;82(2):223-232. doi: 10.1016/j.eururo.2022.03.010.

Cruz F, et al, J Med Econ. 2023 Jan-Dec;26(1):200-207. doi: 10.1080/13696998.2023.2165366

Thank you for your suggestion; I added it to my paragraph.

In section 2:

 Quote Kennelly et al, Eur Urol. . 2022 Aug;82(2):223-232. doi: 10.1016/j.eururo.2022.03.010.

 The approval of 200 U Botox is not only in the US, please correct the sentence

 Describe the CONTENT trial and the comparison between the two toxins from Cruz F, et al,  J Med Econ. 2023 Jan-Dec;26(1):200-207. doi: 10.1080/13696998.2023.2165366

Thank you for your suggestion; I added it to my paragraph.

In page 3, About trigonal injections in NDO patients please comment the paper:

Chen et al, J Spinal Cord Med. 2020

 Thank you for your suggestion; I added it to my paragraph.

Concerning urethral injections:

Describe continence and pad usage. The study quoted is small and how many  patienst were dry? Please describe and make a comment. Also describe the duration of effect of urethral injections.

The study didn’t mention how many patients were dry. Urethral injection mainly focused on the voiding function. Most studies only mentioned about the symptoms improvement.

 In section 4 when addressing ideal doses make clear that doses different than 200 for Botox are not licenced. Also leave clear that doses below 100 U are not effective in NDO/SCI patients.  Refer to  Apostolidis World J Urol. 2013

 Thank you for your suggestion; I added it to my paragraph.

In section 5, comment the duration of effect of BoNT/A injections in the sphincter in the text and in Table 1

 Thank you for your suggestion; I added it to my paragraph.

 In section 8: It is important to refer the maximal  doses of OnabotA and AbobotA in the paediatric population. And to state the best dose in the Austin study (ref 74)

 Thank you for your suggestion; I added it to my paragraph.

State if there is any study with sphincter injection in NDO children

I only found literatures about combined detrusor and sphincter injection. I didn’t find sphincter injection only in NDO children.

Round 2

Reviewer 1 Report

Here are some additional comments to the revised manuscript:

1. 

  1. Line 144 – SCI patients do not normally have spontaneous voiding function – and this sentence needs editing since in its current form, its meaning is very unclear.

I can’t find this sentence. Would you please remind me where the sentence is?

It in in Line 144 - Line 146 in the original version and Line 152 -154 in the revised version. 

Therefore, a lower dose (200 U over 300 U) of BoNT-A (is? )competent enough (sufficient) to produce similar improvements in urinary incontinence and urgency episodes without obviously altering the spontaneous voiding function is required [37,38].

This sentence needs to be revised. Some suggestions are noted above. In addition, most SCI patients that are considering or receiving BoNT-A injections do not have spontaneous voiding function. 

2. The authors should add that how prevalent "abdominal tapping" is to this paragraph (section 4; the first paragraph)

Other comments have been addressed sufficiently. 

Author Response

  1. Line 144 – SCI patients do not normally have spontaneous voiding function – and this sentence needs editing since in its current form, its meaning is very unclear.

 I can’t find this sentence. Would you please remind me where the sentence is?

It in in Line 144 - Line 146 in the original version and Line 152 -154 in the revised version. 

Did you mean “SCI patients commonly have both voiding and storage symptoms”

My meaning is the two symptoms commonly occurred at the same time. NDO with DSD.

Thank you for your suggestion.

Therefore, a lower dose (200 U over 300 U) of BoNT-A (is? )competent enough (sufficient) to produce similar improvements in urinary incontinence and urgency episodes without obviously altering the spontaneous voiding function is required [37,38].

This sentence needs to be revised. Some suggestions are noted above. In addition, most SCI patients that are considering or receiving BoNT-A injections do not have spontaneous voiding function. 

Thank you for your suggestion.

I have revised it

  1. The authors should add that how prevalent "abdominal tapping" is to this paragraph (section 4; the first paragraph)

Thank you for your suggestion.

I have revised it

Other comments have been addressed sufficiently. 

Reviewer 3 Report

The references are out of order.

Ref 76 is the same of ref 11. do not duplicate references

Introduce in introduction when comparing doses between Ona e AbobotA the reference:

Kennelly et al, Eur Urol. . 2022 Aug;82(2):223-232. doi: 10.1016/j.eururo.2022.03.010. 

Author Response

The references are out of order.

Ref 76 is the same of ref 11. do not duplicate references

Introduce in introduction when comparing doses between Ona e AbobotA the reference:

Kennelly et al, Eur Urol. . 2022 Aug;82(2):223-232. doi: 10.1016/j.eururo.2022.03.010. 

Thank you for your suggestion.

I have revised it.
